# Comparison of the Efficacy of *Trichoderma* and *Bacillus* Strains and Commercial Biocontrol Products against Grapevine *Botryosphaeria* Dieback Pathogens

Natalia Langa-Lomba [1,2], Vicente González-García [1], M. Eugenia Venturini-Crespo [3], José Casanova-Gascón [4], Juan J. Barriuso-Vargas [4] and Pablo Martín-Ramos [2,5,*]

1   Department of Agricultural, Forest and Environmental Systems, Agrifood Research and Technology Centre of Aragón, Instituto Agroalimentario de Aragón—IA2 (Universidad de Zaragoza-CITA), Avda. Montañana 930, 50059 Zaragoza, Spain
2   Instituto Universitario de Investigación en Ciencias Ambientales de Aragón (IUCA), EPS, University of Zaragoza, Carretera de Cuarte s/n, 22071 Huesca, Spain
3   Plant Food Research Group, Instituto Agroalimentario de Aragón—IA2 (Universidad de Zaragoza-CITA), Miguel Servet 177, 50013 Zaragoza, Spain
4   Instituto Agroalimentario de Aragón-IA2 (Universidad de Zaragoza-CITA), Avda. Montañana 930, 50059 Zaragoza, Spain
5   Department of Agricultural and Forestry Engineering, ETSIIAA, University of Valladolid, Avenida de Madrid 44, 34004 Palencia, Spain
*   Correspondence: pmr@unizar.es

**Abstract:** Grapevine trunk diseases (GTDs) cause significant yield losses worldwide and limit the lifespan of vineyards. In the last few years, using biological control agents (BCAs) for pruning wound protection has become a promising management strategy for the control of these pathologies. This study aimed to compare the antifungal activities of a grapevine-native *Trichoderma harzianum* isolate and a high-potential *Bacillus velezensis* strain against two pathogenic *Botryosphaeriaceae* species in artificially inoculated, potted, grafted plants under controlled greenhouse conditions, taking three commercial biocontrol products (based on *T. atroviride* I-1237, *T. harzianum* T-22, and *Bacillus subtilis* BS03 strains) as a reference. To reproduce certain field conditions more realistically, inoculation of the protective agents and the pathogens was conducted simultaneously immediately after pruning instead of allowing the BCAs to colonize the wounds before pathogen inoculation. Significant differences in necrosis lengths were detected for both *Neofusicoccum parvum*- and *Diplodia seriata*-infected plants, and a remarkable protective effect of *Bacillus velezensis* BUZ-14 was observed in all cases. *Trichoderma*-based treatments showed different efficacies against the two pathogenic fungi. While the three tested BCAs resulted in significant reductions in vascular necrosis caused by *N. parvum*, they did not significantly reduce *D. seriata* infection compared to the untreated inoculated control. The *B. subtilis* strain was not effective. The reported results provide support for the potential *Bacillus velezensis* may have for pruning wound protection against *Botryosphaeriaceae* fungi, encouraging its evaluation under natural field conditions.

**Keywords:** *Bacillus subtilis*; *Bacillus velezensis*; BCAs; *Diplodia seriata*; GTDs; *Neofusicoccum parvum*; *Trichoderma harzianum*; *Trichoderma atroviride*; *Vitis vinifera*; wound protection

## 1. Introduction

Grapevine trunk diseases (GTDs) are diseases caused by several fungal genera and species that alter wood, causing general decay of the plant and leading to yield reductions and reduced lifespan [1]. Black-foot (mainly caused by species of the genera *Ilyonectria*, *Dactylonectria*, *Campylocarpon*, and *Cylindrocarpon*), Petri, and *Botryosphaeriaceae* dieback (caused by members of the genera *Botryosphaeria*, *Diplodia*, *Lasiodiplodia*, *Neofusicoccum*, and *Dothiorella*) diseases are most commonly associated with young vineyards, while Esca

(caused by species of the genera *Fomitiporia*, *Stereum*, *Inonotus*, *Phaeomoniella*, and *Phaeoacremonium*), eutypiosis (with the genus *Eutypa* as the main causal agent), and black dead arm diseases stand out for adult plants. In the last three decades, these fungal pathologies have become a major concern among winegrowers as they are causing important economic losses worldwide, with an annual cost associated with the replacement of grapevine plants estimated at more than EUR 1.1 billion in 2017 [2].

Interest in the development of new control methods to manage these diseases has increased for a variety of reasons, including the absence of curative crop protection products, the higher impact of these diseases as a result of more intensive vineyard management, and the banning of numerous fungicides of chemical origin. Hence, the lack of options for the control and management of these diseases makes the use of preventive measures, such as pruning wound-protectant biological control agents (BCAs), a key and environmentally friendly strategy [3].

Previous studies exploring the suitability and potential of biological control methods for GTDs have covered both bacterial antagonists (belonging to the genera *Bacillus*, *Pseudomonas*, *Streptomyces*, and *Enterobacter*) [4] and fungal BCAs (including several species of *Fusarium*, *Trichoderma*, and *Epicoccum*, among others) [1], but *Trichoderma* spp. and *Bacillus* spp. are by far the most widely tested and effective microorganisms against wood diseases in grapevines.

*Trichoderma* spp., one of the most widely used microorganisms in integrated pest management [5,6] and extensively licensed and employed as commercial preparations, exhibits complex mechanisms of interest for disease control, such as its hyperparasitic behavior and the production of lytic enzymes, antimicrobial substances, and other secondary metabolites with germicidal action [7], and it has also been reported to be a plant growth promoter [8]. In turn, the antibiosis mechanism based on *Bacillus* species functions through beneficial molecules (including hydrosoluble and volatile metabolites) that induce or trigger plant defense pathways (phytohormone precursors, lipopolysaccharides, siderophores, etc.) [4].

Concerning their efficacy against *Botryosphaeriaceae* dieback (one of the most significant emergent GTDs caused by fungal species belonging to the genera *Botryosphaeria*, *Diplodia*, *Neofussicoccum*, *Lasiodiplodia*, *Dothiorella*, and *Spencermartinsia* [9])—and, in particular, against two of the most frequently isolated species of the group (viz. *Neofusicoccum parvum* (Pennycook & Samuels) Crous, Slippers & A.J.L. Phillips and *Diplodia seriata* De Not)—several *Trichoderma atroviride* P. Karst. And *Trichoderma harzianum* Rifai strains have been reported to reduce these infections [4,8,10–13], albeit with variable results in terms of efficacy, as is also the case for *Bacillus subtilis* (Ehrenberg 1835) Cohn 1872 [4,10,14]. However, with a few exceptions [8,15], many of these BCAs are not native to grapevine plants, and commercial products were developed to control different pathogens in crops other than grapevines. As noted by Pollard-Flamand et al. [8], adopting a BCA product from another climate or ecosystem may present problems as the effectiveness of BCA-based formulations can vary between in vitro and in situ studies performed in different hosts and under different environmental conditions. Consequently, in recent years, there has been a growing interest in conservation biological control (CBC) [16] and the evaluation of the potential of locally isolated endophytic BCAs against GTD fungi, given that they could be better adapted.

Accordingly, a native grapevine strain of *T. harzianum* was chosen from among a set of isolates of this species tested in a previous study [15] due to the good protective results obtained. Thus, the work presented herein aimed to evaluate the efficacy of this microorganism in potted, grafted plants artificially inoculated with *N. parvum* and *D. seriata*, comparing it with that of the promising research-grade strain of *Bacillus velezensis* Ruiz-Garcia et al. 2005 (formerly named *B. amyloliquefaciens* BUZ-14, native to another plant host in the same geographical area) and three commercial biocontrol products (based on *T. atroviride* I-1237, *T. harzianum* T-22, and *Bacillus subtilis* BS03).

## 2. Material and Methods

### 2.1. Plant Material

The two-year-old grafted plants used in the bioassays were supplied by VCR Vivai Cooperativi Rauscedo (Italy) with supplier ID IT-06-1031. The chosen clone was "Tempranillo RJ 43" and the rootstock was "110R VCR114".

### 2.2. Fungal Isolates

The *Botryosphaeriaceae* fungi selected for the assays were a *Neofusicoccum parvum* strain (isolate MYC-1270) and a *Diplodia seriata* strain (isolate MYC-1569) isolated from diseased young (5–7 years old) Aragonese grapevine plants preserved in the fungal living collection of the Mycology Laboratory at the Department of Agricultural, Forestry and Environmental Systems of the Centro de Investigación y Tecnología Agroalimentaria de Aragón (CITA, Zaragoza, Spain). The two pathogenic species were isolated from diseased grapevine plants sampled in wine-producing areas of Aragon (northeast Spain) and characterized using both morphological and molecular methods. In this way, ribosomal ITS sequences of both strains were obtained, and their taxonomical assignment was confirmed by comparison with public databases with the BLASTn tool. The isolates were recovered from cryovials with 20% glycerol at a temperature of $-80\ ^\circ$C as potato dextrose agar (PDA, purchased from Becton, Dickinson, and Company; Franklin Lakes, NJ, USA) subcultures, performing periodic replicates to maintain optimal colonies. These two taxa were selected because they are among the more virulent, polyphagous, and faster-growing *Botryosphaeriaceae*, utilizing larger carbon and nitrogen sources than other species [17].

### 2.3. Treatments

Five biocontrol agents were tested against both pathogens: a native *Trichoderma harzianum* (isolate MYC-V102) strain isolated as an endophyte of grapevine plants originating from healthy samples from vineyards in Aragon and identified at the morphological and molecular level (through BLASTn comparison of its ribosomal ITS sequence) in previous work [15]; a *Bacillus velezensis* strain (BUZ-14) obtained from the Plant Food Research Group Collection at the University of Zaragoza, isolated from the surface of peach fruit from an orchard in Zaragoza, that had been successfully tested for antifungal potential against other phytopatogens [18,19]; the commercial wound protectant Trianum-P® based on *T. harzianum* (strain T-22), developed by Koppert BV (Berkel en Rodenrijs, the Netherlands) and commercialized by Kopert España (La Mojonera, Almería); the commercial wound protectant product Esquive® based on *Trichoderma atroviride* (strain I-1237), developed by Agrauxine S.A. (Quimper, France) and commercialized by Idai Nature S.L. (Valencia, Spain); and the commercial formulation FUNGISEI® based on *Bacillus subtilis* (strain BS03), developed by Seipasa (Valencia, Spain). The *Bacillus velezensis* (strain BUZ-14) had been previously characterized and its evolutionary relationships were elucidated through a phylogenetic reconstruction using Bayesian inference from a comparison of its ribosomal 16S sequence (Figure A1).

### 2.4. Production of T. harzianum and B. velezensis Treatments

To obtain conidial solutions of the native *T. harzianum* strain employed, it was inoculated in sextuplicate (4 mm diameter agar plugs) on PDA plates (12 cm in diameter) and incubated at 25 $^\circ$C in the dark. To harvest the conidia, sterile bidistilled water was poured into each plate, completely covering the colony, and the plates were sealed with Parafilm™. The plates were then shaken to detach the conidia, and the aqueous solution containing the spores was recovered. Subsequently, the conidial solutions were titrated and adjusted using a hematocytometer to obtain a final concentration of $1 \times 10^7$ conidia·mL$^{-1}$. The inoculum was stored in cold storage until subsequent use.

To prepare a fresh cell suspension of the *B. velezensis* BUZ-14 strain, a 24 h old culture on tryptose soy agar (TSA, purchased from Becton, Dickinson, and Company) was transferred to 7 mL of tryptose soy broth (TSB, also supplied by Becton, Dickinson, and Company),



the suspension was incubated at 30 °C for 24 h on a rotary shaker at 150 rpm, and the concentration was finally adjusted to $1 \times 10^7$ CFU·mL$^{-1}$.

### 2.5. Greenhouse Bioassays on Grafted Plants

For the in vivo tests, 232 grafted grapevine plants were used: 100 were infected with *N. parvum* (20 plants/treatment), l00 were infected with *D. seriata* (20 plants/treatment), 20 were used as negative controls (4 plants/treatment), and 12 were used as positive controls (6 plants/pathogen) (Table 1).

**Table 1.** Treatments, concentrations, and replicates used in the bioassays.

| Treatment | Concentration | Pathogen | Number of Replicates |
|---|---|---|---|
| Native *Trichoderma harzianum* | $1 \times 10^7$ conidia·mL$^{-1}$ | *D. seriata*<br>*N. parvum*<br>Negative control | 20<br>20<br>4 |
| *Trichoderma harzianum* T-22 (Trianum-P®) | $1 \times 10^7$ conidia·mL$^{-1}$ | *D. seriata*<br>*N. parvum*<br>Negative control | 20<br>20<br>4 |
| *Trichoderma atroviride* I-1237 (Esquive®) | $1 \times 10^7$ conidia·mL$^{-1}$ | *D. seriata*<br>*N. parvum*<br>Negative control | 20<br>20<br>4 |
| *Bacillus velezensis* BUZ-14 | $1 \times 10^7$ CFU·mL$^{-1}$ | *D. seriata*<br>*N. parvum*<br>*Negative control* | 20<br>20<br>4 |
| *Bacillus subtilis* BS03 (FUNGISEI®) | $1 \times 10^7$ CFU·mL$^{-1}$ | *D. seriata*<br>*N. parvum*<br>*Negative control* | 20<br>20<br>4 |
| - | - | *D. seriata* positive control<br>*N. parvum* positive control | 6<br>6 |

Each grafted plant was grown in a 3.5 L plastic pot with a mixed substrate of peat and natural grapevine soil (75:25) with a loamy texture from an experimental vineyard in the natural region "Hoya de Huesca" (Huesca, NE Spain) and treated in an autoclave, incorporating a slow-release fertilizer when necessary throughout the study period. Grapevine plants were kept in a greenhouse with drip irrigation and an anti-weed net at the Escuela Politécnica Superior, Universidad de Zaragoza, for six months (from May to November 2022). The cooling system installed in the greenhouse controlled parameters such as ventilation, humidity, and temperature. The mean temperature during the experiment ranged from 10 to 29 °C (day/night), while the relative humidity (RH) varied over the interval of 30–45%.

Rootstocks were simultaneously inoculated in May 2022 with the five BCAs and the two pathogens (*N. parvum* and *D. seriata*). Inoculations were performed on the rootstock trunk at two points below the grafting point. Slits (15 mm in diameter and 5 mm deep) were made with a scalpel. The protective treatments were applied in different ways on the slits. For native and commercial *T. harzianum* strains, the inocula were applied using alginate beads as a carrier, prepared by dispersing fungal propagule solutions in a 3% sodium alginate solution in a 1:4 ratio (i.e., 20 mL treatment/80 mL sodium alginate). Once the mixture was homogenized, the solution was dispensed dropwise over a 3% calcium carbonate solution to produce the ionic exchange and spherify the resulting solution. As a result, beads with ∅ = 0.4–0.6 cm containing the different treatments were obtained. Two beads, one on each side of the agar plug with the pathogen, were placed on each wound (Figure 1a). The commercial *T. atroviride* I-1237 treatment was applied as a spray to the wound and allowed to dry (Figure 1b). The treatment with *Bacillus velezensis* BUZ-14 was amended with 1% ALKIR® wetting agent (De Sangosse Ibérica, Valencia, Spain),

applied to each wound using a pipette (1.5 mL per wound), and allowed to dry (Figure 1c). The same procedure was followed for the commercial formulation of *Bacillus subtilis* (BS03). Agar plugs (5 mm in diameter) from fresh pathogen PDA cultures were placed on the center of the slit, and the wound was covered with absorbent sterile cotton moistened with sterile bidistilled water and sealed with Parafilm™.

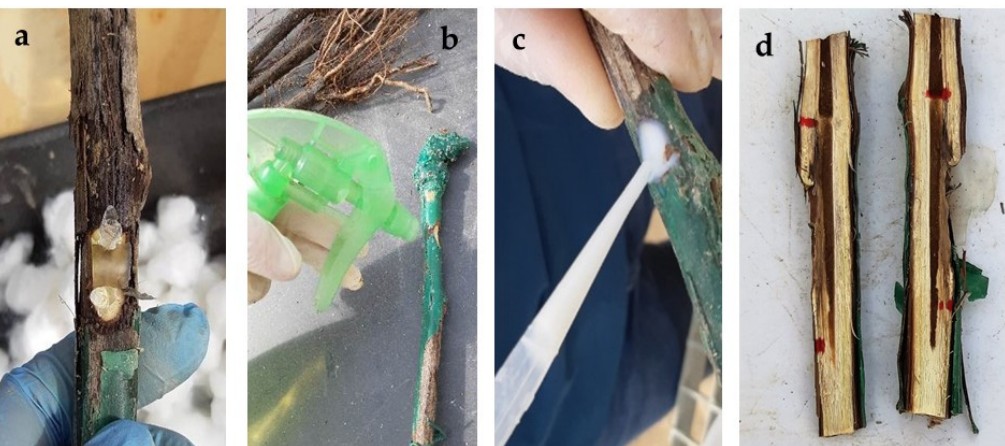

**Figure 1.** BCA application procedures. (**a**) Treatment with *T. harzianum* inocula dispersed in alginate beds placed at both sides of the agar plug together with the pathogen; (**b**) spraying of *T. atroviride* inocula; (**c**) application of *Bacillus* spp.-based treatments using a pipette; (**d**) longitudinally opened plant showing vascular necroses from the inoculation points.

In November 2022, the grafted plants were cut into sections and opened longitudinally, and the lengths of the vascular necroses were evaluated (Figure 1d). Lesions were measured longitudinally on both sides of each inoculation point in the upper and lower directions, taking the average of the four measurements as the necrosis length for each inoculation point. Finally, the two mentioned pathogens were re-isolated directly from the vascular lesions and morphologically identified to fulfill Koch's postulates.

### 2.6. Statistical Analyses

Since the normality and homoscedasticity requirements were not met, the Kruskal–Wallis nonparametric test was used, with the Conover–Iman test employed for post hoc multiple pairwise comparisons. R statistical software was used for all of the statistical analyses [20].

## 3. Results

### 3.1. Comparison of Efficacies against N. parvum

None of the biocontrol agents tested fully inhibited the vascular symptoms of *N. parvum* (Figure 2). However, some of the treatments were effective in reducing the length of the necrosis produced—with statistically significant differences ($p$-value < 0.0001)—in comparison to the controls inoculated only with the pathogen. As shown in Table 2, *B. velezensis* (BUZ-14) was found to be the most effective treatment, with an efficacy comparable to that of the commercial *T. atroviride* formulation. The native strain of *T. harzianum* showed an intermediate efficacy, comparable to that of the commercial T-22 strain, with vascular necrosis lengths that were also significantly different from those of the positive (pathogen) control. Concerning the *B. subtilis*-based product, the necrosis lengths were comparable to those of the positive control.

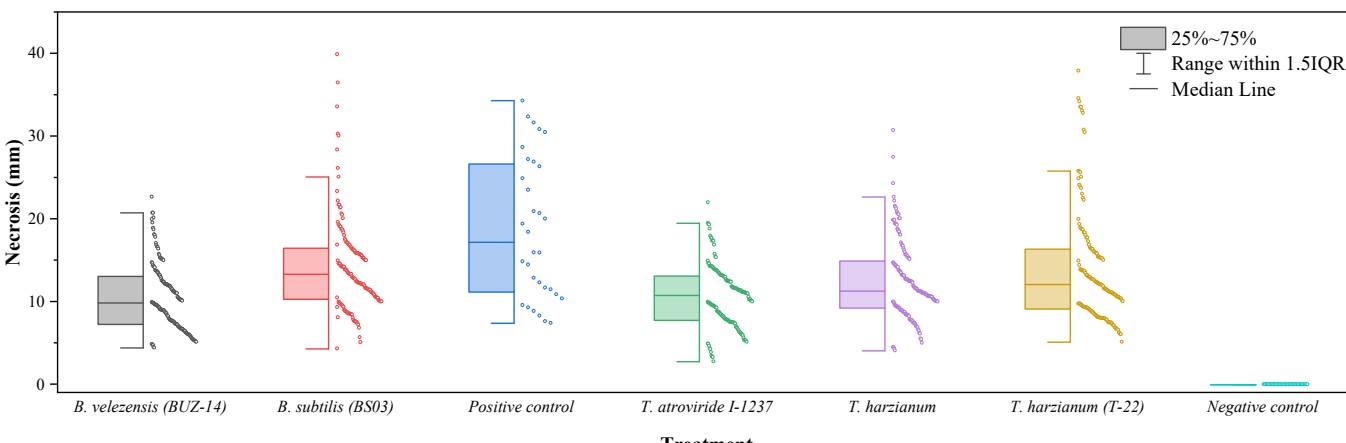

**Figure 2.** Box-plot of vascular necrosis lengths for *N. parvum*.

**Table 2.** Kruskal–Wallis test and multiple pairwise comparisons using the Conover–Iman procedure for the lengths of the vascular necroses for *N. parvum*.

| Treatment | Mean of Ranks | Groups | | | |
|---|---|---|---|---|---|
| Negative control | 17.000 | A | | | |
| *B. velezensis* (BUZ-14) | 360.994 | | B | | |
| *T. atroviride* (I-1237) | 372.791 | | B | | |
| *T. harzianum* (native) | 458.920 | | | C | |
| *T. harzianum* (T-22) | 487.763 | | | C | |
| *B. subtilis* (BS03) | 533.659 | | | C | D |
| *N. parvum* positive control | 623.578 | | | | D |

*3.2. Comparison of Efficacies against D. seriata*

Concerning the efficacy of the treatments against *D. seriata* (Figure 3), statistically significant differences (*p*-value < 0.0001) were also detected depending on the treatment considered (Table 3). *Bacillus velezensis* (BUZ-14) was again the treatment with the highest efficacy and the only one for which necrosis lengths significantly differed from those of the positive (pathogen) control. The three *Trichoderma* strains did not control the growth of *D. seriata* in a significant manner, and necrosis lengths larger than those of the positive control were observed for the *B. subtilis* treatment.

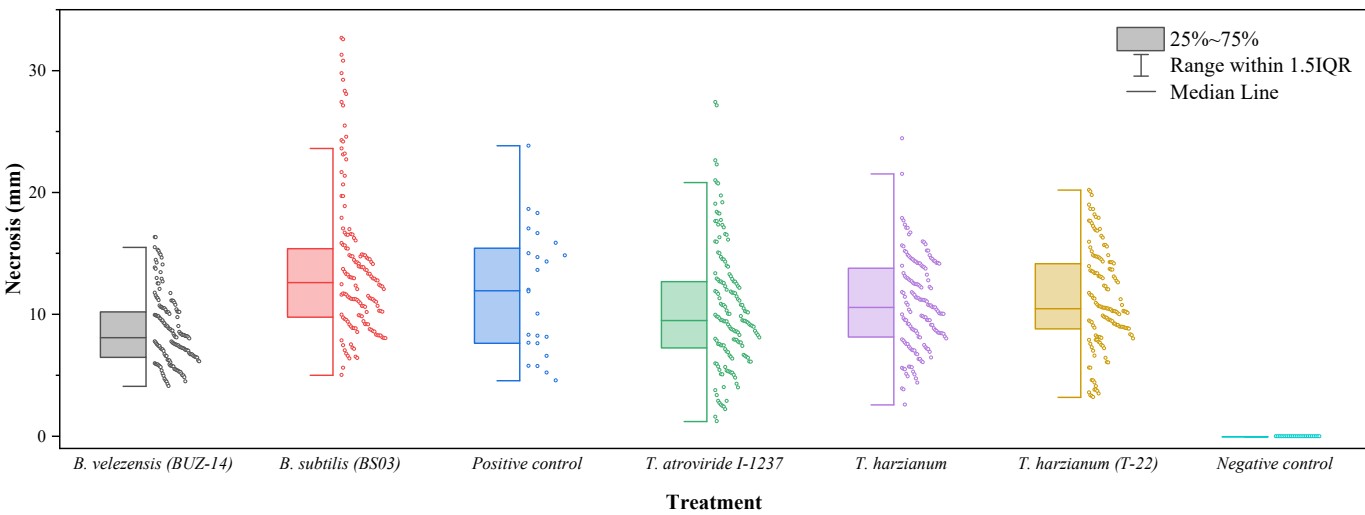

**Figure 3.** Box-plot of vascular necrosis lengths for *D. seriata*.

**Table 3.** Kruskal–Wallis test and multiple pairwise comparisons using the Conover–Iman procedure for the lengths of the vascular necroses for *D. seriata*.

| Treatment | Mean of Ranks | Groups | | | |
|---|---|---|---|---|---|
| Negative control | 12.500 | A | | | |
| *B. velezensis* (BUZ-14) | 297.784 | | B | | |
| *T. atroviride* (I-1237) | 399.234 | | | C | |
| *T. harzianum* (native) | 438.197 | | | C | |
| *T. harzianum* (T-22) | 457.480 | | | C | |
| *D. seriata* positive control | 470.813 | | | C | D |
| *B. subtilis* (BS03) | 548.569 | | | | D |

## 4. Discussion

The present work explored the potential of several native fungal and bacterial microorganisms as microbial antagonists against two of the most important pathogens associated with the "*Botryosphaeria* dieback" disease in grapevine. Their potential was tested in comparison to other commercial preparations based on similar microorganisms. When comparing our results with other BCA-based studies with similar GTD fungi—given that the application methods, product concentrations, testing conditions, durations of the assays, etc. differed from one study to another—the efficacy comparisons presented below should be treated with caution.

The strong and remarkable antifungal activity observed for *B. velezensis* BUZ-14 against both pathogens was consistent with previous findings reported by Calvo et al. [18,19] with other important pathogens, such as *Botrytis cinerea* Pers.; *Monilinia fructicola* (G. Winter) Honey; and *Monilinia laxa* (Aderh. & Ruhland) Honey. In these studies, the authors observed the antifungal activity of *B. velezensis* in direct applications on grape and peach fruits, and it was significantly increased when these treatments consisted of culture cell-free supernatant containing hydrosoluble and volatile metabolites. Other *B. amyloliquefaciens* strains have also been reported to be effective against grapevine fungal pathogens. For instance, Alfonzo et al. [21] reported that the crude protein extract obtained from the culture supernatant of a grapevine-native strain of *B. amyloliquefaciens* was effective in vitro against certain grapevine-associated fungi, including GTD pathogens, such as *Fomitiporia mediterranea* M. Fisch., *Lasiodiplodia theobromae* (Pat.) Griffon & Maubl., *Phaeoacremonium aleophilum* W. Gams, Crous, M.J. Wingf. & Mugnai, and *Phaeomoniella chlamydospora* (W. Gams, Crous, M.J. Wingf. & L. Mugnai) Crous & W. Gams. On the other hand, Brown et al. [22] reported a low in vivo efficacy for *B. amyloliquefaciens* against a set of GTD-related fungi that included *N. parvum*, suggesting that this BCA should be applied before the pathogenic infection as a protective strategy to allow the BCA to establish itself and be active in pruning wounds before being challenged by the pathogen. It should be noted that these authors used a formulation based on a wettable powder consisting of CFUs of the antagonist, the activation dynamics (and effectiveness) of which should have been slower than if a solution directly containing secondary metabolites was applied.

Concerning the low activity observed for the *B. subtilis*-based formulation, it should be taken into account that RADISEI® is not commercialized as a wound protectant but as a biostimulant. Nonetheless, it should be noted that there are mixed results about its efficacy in the literature. For instance, Halleen et al. [23] referred to the fact that it was not effective at all against *Eutypa lata* (Pers.) Tul. & C. Tul., and Kotze et al. [10] reported low efficacies against both *Diaporthe ampelina* (Berk. & M.A. Curtis) R.R. Gomes, C. Glienke & Crous and *N. parvum.* On the other hand, *B. subtilis* PTA-271—alone [24] or in combination with *T. atroviride* SC1 [4]—has been shown to be effective against *N. parvum* Bt67. According to Rezgui et al. [14], *B. subtilis* B6 had a positive effect on young vines of the "Muscat d'Italie" cultivar, reducing the size of the wood necrosis caused by *N. parvum*. Rusin et al. [25] also found that *B. subtilis* reduced the severity of *L. theobromae* after winter pruning in cv. "Syrah" grapevines. Alfonzo et al. [26] confirmed—in vitro—that *B. subtilis* AG1 showed antagonistic behavior against *P. aleophilum*, *P. chlamydospora*, and *Botryosphaeria rhodina* (Berk. & M.

A. Curtis) Arx, a result consistent with those reported by Sebestyen et al. [27] for *B. subtilis* (and *T. atroviride*)—also in vitro—against *E. lata*, *P. minimum*, and *P. chlamydospora*. In all these studies, regardless of whether *B. subtilis* was applied as a cell suspension of known concentration or as a crude extract of metabolites (CME), the designs of the treatments usually had in common that they were all employed as pruning wound protectors, where the existence of a protective effect depends on the early colonization by the antagonist prior to pathogenic infection. Thus, a possible explanation for the results obtained here is that *B. subtilis* reduces the incidence of GTD pathogens compared to untreated controls when applied in wounds several days before infection [10].

In the case of fungal BCAs, studies also report varying results on the efficacy of *Trichoderma* spp. against GTD fungi. *T. harzianum* has been reported to reduce the growth of *E. lata* in vitro [28]. In vivo, Rusin et al. [25] indicated that it was effective against *L. theobromae* in "Syrah" grapevines in terms of decreasing its re-isolation rates after treatments. Di Marco et al. [29] assayed *T. harzianum* T39 (Trichodex®) against *P. chlamydospora* in grafted, potted vines, observing that its application prevented black goo and necrosis in the wood below the wound. John et al. [30] found that *T. harzianum* applied to grapevine pruning wounds as a spore suspension reduced the recovery of *E. lata* both in the glasshouse and in the field but noted that—in field experiments—the incorporation of the *Trichoderma* formulate before the pathogen reduced the recovery of the latter. Other *Trichoderma harzianum*-based products also protected pruning wounds in "Cabernet Sauvignon", "Sauvignon blanc", "Red Globe", and "Bonheur" grapevine cultivars, reducing the incidence of *E. lata* and other GTD pathogens [31].

Regarding previous results for *T. atroviride*, in an in vitro screening of *Trichoderma* isolates for the biocontrol of black foot disease pathogens, van Jaarsveld et al. [32] found that two isolates of *T. atroviride* showed the highest overall mycelium growth inhibition (although the efficacy was isolate-dependent, both for *Trichoderma* spp. and the pathogen). Commercial products based on *Trichoderma atroviride* significantly reduced pruning wound infection by GTD fungi, including *N. parvum* (by 80%) and *D. seriata* (by 85%), in studies conducted in South Africa [10]. Pintos et al. [33] also showed that the treatment of pruning wounds with a commercial product based on *T. atroviride* resulted in reductions in the recovery and necrosis lengths of *Botryosphaeriaceae* spp. by 65.7% to 91.9%. Urbez-Torres et al. [11], in a study that evaluated the potential of a collection of strains of different Italian *Trichoderma* species for use as pruning wound protectors, found that a *T. atroviride* isolate effectively protected pruning wounds in detached cane assays against *D. seriata* and *N. parvum* for at least 21 days after treatment. In another work conducted with native grapevine *Trichoderma* isolates from British Columbia (Canada) [8], it was reported that—in in planta detached cane assays under controlled greenhouse conditions—one isolate of *T. atroviride* provided 93% to 100% pruning wound protection against *D. seriata* and *N. parvum* for up to 21 days after treatment, respectively, provided that these two *Botryosphaeriaceae* fungi were inoculated at least 24 h after the protective treatment. Strong protection of pruning wounds against *E. lata* and *N. parvum* was reported by Blundell et al. [12], employing the biofungicide Vintec® based on *T. atroviride*. The same commercial product tested here (Esquive®) was effective in the control of *L. theobromae* on greenhouse-kept grapevines of cv. "Cabernet Sauvignon" and cv. "Touriga Nacional" [34] alone and in combination with LC2017, which is a low-copper-based product with an elicitor effect. Other studies based on *T. atroviride* SC1 (the microorganism formulated in the commercial product Vintec®) showed promise in both reducing infections during the grafting process [7] and protecting pruning wounds in field experiments [35]. Conversely, *T. atroviride*-based formulations did not reduce infection by *D. seriata* or *P. chlamydospora* compared to the untreated inoculated control in field trials conducted in Spain, even though the pathogens were artificially inoculated on the grapevine plants [13]. A tentative explanation for these inconsistencies found in the literature could be the non-optimized application time, given that, for *T. atroviride* or *T. harzianum* in vines at the breaking of dormancy, colonization has been shown [36] to be highest at 6 and 24 h after application.

Additionally, the effect of the plant genotype cannot be excluded, and some authors have reported that the wound protection effect of *Trichoderma* spp. is dependent on the grapevine cultivar [37]. As a rule, most studies evaluating *Trichoderma*-based products typically delay inoculation of pruning wounds with GTD fungi for up to 7 days after treatment to give the product a certain advantage in establishing itself and colonizing exposed wood surfaces after pruning. However, this approach could be controversial, as pruning wounds can be infected immediately after pruning if spores are present in the environment, especially in grapevine management systems or bioclimatic situations where the pruning season may coincide with the production of infective primary inoculum (both sexual and asexual propagules) [3], which supports the procedure chosen for the assays presented here.

Taking a look at the mode of action of the assayed BCAs, the efficacy of *B. velezensis* (BUZ-14) should be attributed to the production of iturin A (a cyclic lipopeptide) [19]. However, Calvo et al. [38] reported that certain volatile organic compounds (VOCs) could also be involved in the fungal growth inhibition mechanism observed for *B. velezensis*. Given the type of experimental design presented here, where the antagonist was formulated with an adjuvant agent and, subsequently, sealed inside the wound, the action of some compounds of the volatilome cannot be ruled out. In turn, the mode of action of *Trichoderma* spp. should be attributed to 6-pentyl-a-pyrone (a major secondary metabolite by quantity that accumulates in the culture filtrates of *T. harzianum* and *T. atroviride*), which has been shown to inhibit mycelial growth and ascospore/conidia germination in *E. lata*, *Neofusicoccum australe* (Slippers, Crous & M.J. Wingfield) Crous, Slippers & A.J.L. Phillips, *N. parvum*, and *P. chlamydospora* [39], together with the widely known mechanisms based on hyperparasitism that are habitually exhibited by members of the genus and a high rate of colonization of the plant surface thanks to their rapid growth.

## 5. Conclusions

In the bioassays conducted on grafted grapevine plants in controlled conditions presented herein, significant differences in the control of vascular necrosis lengths with different BCA-based products (including native microorganisms and those under experimental development) were found both for *N. parvum*- and *D. seriata*-infected plants. While treatments with *T. harzianum* and *T. atroviride* only resulted in significant reductions in necrosis caused by *N. parvum*, a remarkable protective effect from *Bacillus velezensis* BUZ-14 was detected against the two etiological agents. Therefore, the reported results call for further research on this promising iturin A-producing biocontrol agent.

**Author Contributions:** Conceptualization, V.G.-G., J.C.-G. and P.M.-R.; methodology, V.G.-G. and J.C.-G.; validation, M.E.V.-C. and J.J.B.-V.; formal analysis, J.C.-G. and P.M.-R.; investigation, N.L.-L., V.G.-G., M.E.V.-C., J.C.-G., J.J.B.-V and P.M.-R.; resources, J.C.-G.; writing—original draft preparation, N.L.-L., V.G.-G., M.E.V.-C., J.C.-G., J.J.B.-V and P.M.-R.; writing—review and editing, N.L.-L., V.G.-G., J.C.-G. and P.M.-R.; supervision, V.G.-G. and P.M.-R. All authors have read and agreed to the published version of the manuscript.

**Funding:** This research was co-financed by the European Union's Connecting Europe Facility (CEF), grant no. INEA/CEF/ICT/A2018/1837816 GRAPEVINE (hiGh peRformAnce comPuting sErvices for preVentIon and coNtrol of pEsts in fruit crops) project.

**Data Availability Statement:** The data presented in this study are available upon request from the corresponding author. The data are not publicly available due to their relevance to an ongoing Ph.D. thesis.

**Acknowledgments:** The authors thank Seipasa for kindly providing the FUNGESEI® product used in the study.

**Conflicts of Interest:** The authors declare no conflict of interest. The funders had no role in the design of the study; in the collection, analyses, or interpretation of data; in the writing of the manuscript; or in the decision to publish the results.

## Appendix A

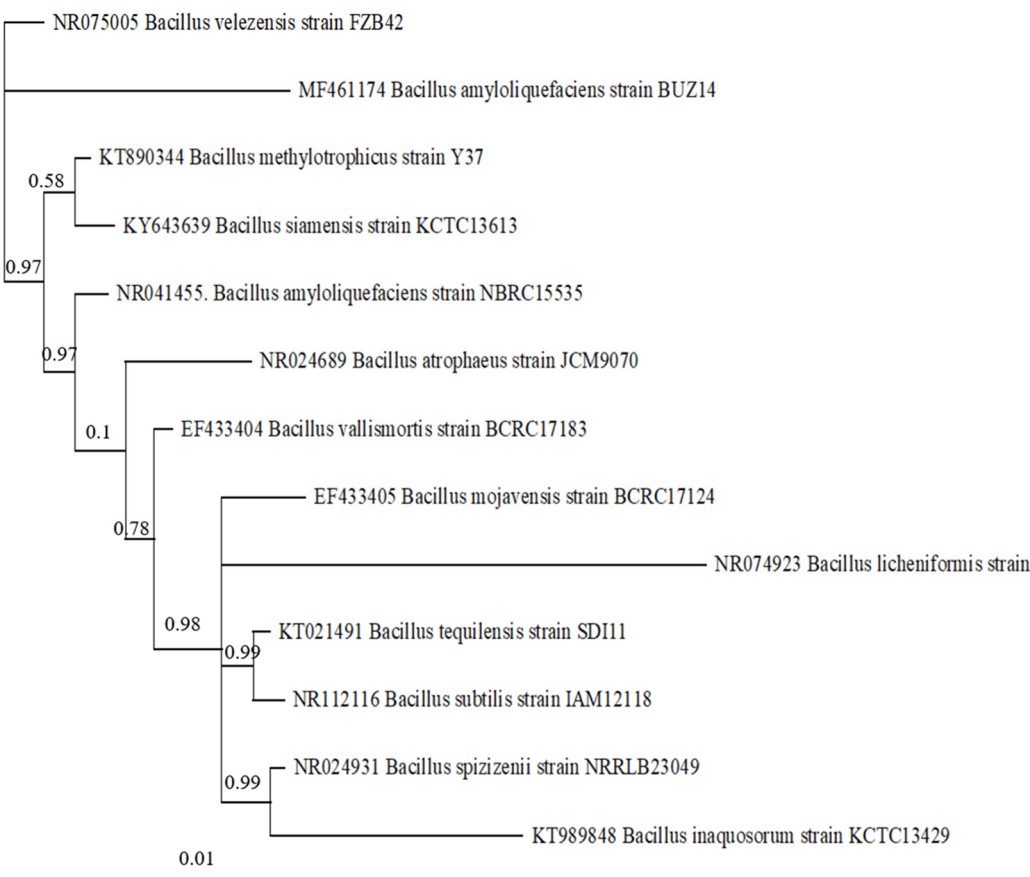

**Figure A1.** The Bayesian 50% majority-rule consensus tree obtained after 2,000,000 generations, inferred from the ribosomal 16S sequence of *B. velezensis* strain BUZ14 and allied *Bacillus* species. The numbers above nodes represent Bayesian posterior probabilities. Only nodes significantly supported bear values. GenBank accession numbers are indicated next to the specific epithets.

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
