# Peer review of "Comparison of the Efficacy of Trichoderma and Bacillus Strains and Commercial Biocontrol Products against Grapevine Botryosphaeria Dieback Pathogens"

_agronomy, doi:10.3390/agronomy13020533_

Round 1

Reviewer 1 Report

The research entitled (Comparison of Efficacy of Novel Trichoderma and Bacillus Strains and Commercial Biocontrol Products against grapevine Botryosphaeria Dieback Pathogens) investigates briefly the efficacy of one isolate of Trichoderma and Bacillus velezensis (previously published) against grafted plants artificiallyinoculated with N. parvum and D. seriata, comparing it with three commercial biocontrol products.

1.      First the research is data in brief, not an article; it constricted the whole experiment to one measured parameter, the lengths of the vascular necrosis.

2.      Paper title includes the word novel, although these isolates have already been utilized against phytopathogens and published in previous papers as written by the authors???

3.      What is the mechanism of controlling the disease here?

4.      The soil type, analysis, and greenhouse conditions are missing.

Author Response

Extensive editing of English language and style required

Response: We have further checked the English during the revision, and the new version of the document has been checked by a Cambridge Proficiency in English (CPE) certificate holder (corresponding to the C2 level of the CEFR [https://www.cambridgeenglish.org/exams-and-tests/proficiency/]).

The research entitled (Comparison of Efficacy of Novel Trichoderma and Bacillus Strains and Commercial Biocontrol Products against grapevine Botryosphaeria Dieback Pathogens) investigates briefly the efficacy of one isolate of Trichoderma and Bacillus velezensis (previously published) against grafted plants artificially‐inoculated with N. parvum and D. seriata, comparing it with three commercial biocontrol products.

Q1. First the research is data in brief, not an article; it constricted the whole experiment to one measured parameter, the lengths of the vascular necrosis.

Response: The document type has been changed to ‘Brief report’, given that ‘Data in brief’ is not one of the article types in Agronomy.

Q2. Paper title includes the word novel, although these isolates have already been utilized against phytopathogens and published in previous papers as written by the authors???

Response: ‘Novel’ has been removed from the title, as suggested by the Reviewer.

Q3. What is the mechanism of controlling the disease here?

Response: In the case of the protective effects observed for the Trichoderma harzianum strain employed here, we propose a complex mechanism of action, based on (1) the ability for rapid colonization of plant tissue (and subsequent blockage thereof) by Trichoderma, (2) the hyperparasitic action of the antagonist, forming appressorium-like structures, papules, etc. and finally lysing the hyphal walls of the pathogen, and (3) the production and secretion of secondary metabolites with germicidal action by the artificially inoculated biological control agent. In the case of B. velezensis, the antagonistic effects should be attributed to iturin A production, together with certain volatile organic compounds (VOCs) that could have been released in the course of the bioassay. Both mechanisms of action are now commented on in the discussion section, as suggested by the Reviewer.

Q4. The soil type, analysis, and greenhouse conditions are missing.

Response: Data on soil type and greenhouse growing parameters have been included in the revised document, as suggested by the Reviewer.

Reviewer 2 Report

The article focuses on using BCAs against two pathogenic Botryosphaeriaceae species causing grapevine trunk diseases. In a green-house experiment authors test various Botrytis and Trichoderma isolates (both native isolates and commercially available products) on artificially inoculated potted plants.  Variable efficacy was achieved for testes microorganisms.
The article is well organised and contains all necessary components.  Sections are developed well enough. The experiment is mostly well designed and methodology is clearly explained. The results are reproducible. The theory is connected to the presented data. The cited literature is relevant, however some more newer findings could be presented. Figures present the methods clearly for the reader to understand.  Tables are easy to read and interpret.
The article is mostly easy to read, however,  language could be improved (perhaps by being checked by a native speaker and/or a professional service) and the article would benefit from being carefully edited.
The results are convincing, however, they are not really novel. They re-assure current state of knowledge rather than advance it. I would still recommend for the to be published since various strains of known beneficial microorganisms need to be tested against plant pathogens. 
The article could be improved by adding details explaining the mode of action for tested treatments. It is mentioned in the Discussion sections, however authors did not conduct such experiment (or at least it is not mentioned in the paper). Did you test for the presence of VOCs or iturin A or secondary metabolites? It would be highly beneficial to add such details, since the green-house experiment is not novel enough on its own.
The article fits the scope of the journal and may be interesting for some readers. It can be considered for publication after some improvements.

Specific comments:
Line 42: remove the so called, usually defined as
Line 43 use disease instead of syndrome
Line 43: which species?
Line 60: citation needed
Line 61: citation needed
Line 65: citation needed
Line 89: rephrase for clarity
Line 104: Can strain codes be added?
Lines 109, 132: PDA manufacturer?
Lines 130-1: rephrase for clarity
Line 139: remove _
Line 140: TSA manufacturer
Lines 146-9: Explain why there were less plants per treatment in positive and negative controls than in other combinations.
Line 164: How much treatment was applied per plant?
Line 171: Was the temperature steady? What was it? Minimum/maximum if variable.
Lines 193-4: Rewrite for clarity.
Table 1 and 2: Why there is no negative control in there since it was mentioned in Methods?
Line 217: Remove so-called.
Line 231: Citation needed.
Lines 243-6: A bit unclear, rephrase, please.

Author Response

The article is mostly easy to read, however, language could be improved (perhaps by being checked by a native speaker and/or a professional service) and the article would benefit from being carefully edited.

Response: We have further checked the English during the revision, and the new version of the main text has been checked by a Cambridge Proficiency in English (CPE) certificate holder (corresponding to the C2 level of the CEFR [https://www.cambridgeenglish.org/exams-and-tests/proficiency/]).

The results are convincing, however, they are not really novel. They re-assure current state of knowledge rather than advance it. I would still recommend for the to be published since various strains of known beneficial microorganisms need to be tested against plant pathogens.

The article could be improved by adding details explaining the mode of action for tested treatments. It is mentioned in the Discussion sections, however authors did not conduct such experiment (or at least it is not mentioned in the paper). Did you test for the presence of VOCs or iturin A or secondary metabolites?

Response: In the case of the native strain of T. harzianum, it was chosen for its good antagonistic capacity, as observed in a previous study [Agronomy 2022, 12, 336]. In that work, we investigated mycelial growth inhibition in plate assays, and the selected strain showed hyperparasitism phenomena against one of the two pathogens tested in this study, N. parvum, producing (see illustrations in the aforementioned article) interactions with the pathogen based on penetration structures, papilla-like bodies, appressoria, lysis of pathogen hyphae, profuse sporulation, intracellular growth, or the presence of typical coil hyphae. In addition, the aforementioned paper also provides insights on the probable mechanism of protection by T. harzianum in the in planta assays, highlighting the ability of the antagonist to protect artificially infected plant tissue when the protective strain was incorporated before the pathogenic species, suggesting a rapid colonization and tissue blockade by Trichoderma prior to pathogen attack. As for B. velezensis, on this occasion, the production capacity of iturin A and other secondary metabolites or the presence of VOCs were not tested. However, the characterization of the profile and antifungal activity of the different secondary metabolites (including iturin A) present in a crude extract of the microorganism, as well as the protective effect of its volatilome, both in vitro and in planta, has already been carried out by some of the co-authors of this work in previous studies [Oeno One 2021, 55, 228-243 and Food Microbiology 2019, 82, 62-69] with the same strain (BUZ-14) of B. velezensis. The possible mechanisms of action attributable to these two native microorganisms used here as biocontrol agents have been included in the revised version of the discussion.

It would be highly beneficial to add such details, since the green-house experiment is not novel enough on its own.

The article fits the scope of the journal and may be interesting for some readers. It can be considered for publication after some improvements.

Specific comments:

Line 42: remove the so called, usually defined as

Response: ‘so called’ and ‘usually defined as’ have been removed.

Line 43 use disease instead of syndrome

Response: ‘syndrome’ has been replaced with ‘diseases’, as suggested by the reviewer.

Line 43: which species?

Response: The most important genera responsible for the cited syndromes are now specified in the manuscript after the syndromes for which they contain etiologic agents. Taking into consideration that, to date, more than 100 fungal species related to GTDs have been identified, for space limitations, we have chosen to include only the main fungal genera involved.

Line 60: citation needed

Response: The manuscript cites a review by Gramaje et al. [Plant disease 2018, 102, 12-39] where an analysis of the management strategies of the different GTDs was carried out, with special emphasis on current methodologies and future perspectives within the framework of integrated pest management.

Line 61: citation needed

Response: According to the Reviewer’s suggestion, the two citations that refer to the use of control methods for wood diseases based on bacterial and fungal antagonists have been reorganized, in such a way that the citation relevant for each type of BCA now appears next to that BCA.

Line 65: citation needed

Response: As suggested by the Reviewer, two citations have been added to the manuscript.

Line 89: rephrase for clarity

Response: As suggested by the Reviewer, the sentence has been rephrased.

Line 104: Can strain codes be added?

Response: Strain codes have been added.

Lines 109, 132: PDA manufacturer?

Response: PDA, TSA, and TSB were purchased from Becton, Dickinson and Company (Franklin Lakes, NJ, USA). This has been clarified in the revised manuscript.

Lines 130-1: rephrase for clarity

Response: As suggested by Reviewer, the sentence has been simplified for better understanding.

Line 139: remove _

Response: ‘_’ has been removed.

Line 140: TSA manufacturer

Response: PDA, TSA, and TSB were purchased from Becton, Dickinson and Company (Franklin Lakes, NJ, USA).

Lines 146-9: Explain why there were less plants per treatment in positive and negative controls than in other combinations.

Response: The existence of a lower number of control plants (positive and negative) was due to an adjustment in the total number of plants available. As it was limited, we decided to reduce the number of controls in such a way that, while being sufficient for statistical analyses, the number of plants subjected to the different treatments would be maximized.

Line 164: How much treatment was applied per plant?

Response: The amount of bacterial formulate per wound has been specified in the manuscript (1.5 mL/wound).

Line 171: Was the temperature steady? What was it? Minimum/maximum if variable.

Response: Greenhouse growing conditions have been specified in the manuscript.

Lines 193-4: Rewrite for clarity.

Response: As suggested by the Reviewer, the sentence has been rewritten for a better understanding of the concept.

Table 1 and 2: Why there is no negative control in there since it was mentioned in Methods?

Response: Tables 2 and 3 have been updated to include the negative control.

Line 217: Remove so-called.

Response: ‘so-called’ has been removed.

Line 231: Citation needed.

Response: In line 233, the work of Alfonzo et al. (2012), where the effectiveness of a strain of B. amyloliquefaciens against a set of grapevine-associated pathogens, including GTDs species, is cited. In addition, a work by Brown et al. (2021) on the effectiveness of another strain of the aforementioned bacterium against N. parvum has also been added a few lines below.

Lines 243-6: A bit unclear, rephrase, please.

Response: As suggested by the Reviewer, the sentence has been rephrased for clarity.

Reviewer 3 Report

Dear authors,

The manuscript Agronomy 2160203 entitle “Comparison of Efficacy of Novel Trichoderma and Bacillus strains and commercial Biocontrol products against grapevine Botryosphaeria Dieback Pathogens” includes the evaluation of effects of two biocontrol agents Trichoderma and Bacillus against a disease caused by Botryosphaeria in grapevine. The manuscript can be considered for publication after the following observations were made.

 Specific comments

1.     In the methodology paragraph the authors need mention must be mentioned how the encapsulation process with alginate was carried out.

2.     In the results paragraph, the authors need show the bioassays analysis in a boxplot graph and show the standard deviations.

Major points

·       The authors need improve discussion paragraph.

Minor points

·       In line 42: Please delete “so-called”.

·       In line 129: Please write the “T.harzianum” and “B. veleznsis”  in italics.

·       In line 191:  Write “N. parvum” in italics, please.

·       In line 204:  Write “D. seriata” in italics.

·       In line 217: Please write “ Botryosphaeria” in italics.

Author Response

The manuscript Agronomy 2160203 entitle “Comparison of Efficacy of Novel Trichoderma and Bacillus strains and commercial Biocontrol products against grapevine Botryosphaeria Dieback Pathogens” includes the evaluation of effects of two biocontrol agents Trichoderma and Bacillus against a disease caused by Botryosphaeria in grapevine. The manuscript can be considered for publication after the following observations were made.

Specific comments

Q1. In the methodology paragraph the authors need mention must be mentioned how the encapsulation process with alginate was carried out.

Response: The procedure described in subsection 2.5 has been expanded to provide further detail on the alginate beads preparation procedure. It now reads: “For native and commercial T. harzianum strains, the inocula were applied using alginate beads as a carrier, prepared by dispersing fungal propagules solutions in a 3% sodium alginate solution in a 1:4 ratio (i.e., 20 mL treatment/80 mL sodium alginate). Once the mixture was homogenized, the solution was dispensed dropwise over a 3% calcium carbonate solution to produce the ionic exchange and spherify the resulting solution. As a result, beads of = 0.4–0.6 cm containing the different treatments were obtained

Q2. In the results paragraph, the authors need show the bioassays analysis in a boxplot graph and show the standard deviations.

Response: Please kindly note that standard deviations are only applicable to parametric statistics (IQR would be used instead for non-parametric statistics, applicable to our data, which does not meet normality and homoscedasticity requirements). Box plot graphs have been included to address this point raised by the Reviewer.

Major points

Q3. The authors need improve discussion paragraph.

Response: As suggested by the Reviewer, the discussion section has been modified in the manuscript, correcting the wording of some sentences, simplifying the exposition of hypotheses, and, in a certain sense, lightening it for a better understanding.

Minor points

Q4. In line 42: Please delete “so-called”.

Response: Deleted.

Q5. In line 129: Please write the “T.harzianum” and “B. veleznsis” in italics.

Response: Given that the journal’s style for subsection headings uses italics, scientific names (which are always italicized) were indicated in plain font according to APA style requirements (APA 7, section 6.22), which indicate that “Use reverse italics when a word or phrase is italicized within a title or other phrase that is already italicized. Reverse italics means that the word or phrase that would normally be italicized is not italicized (presented in plain text)” [https://apastyle.apa.org/style-grammar-guidelines/italics-quotations/italics]

Q6. In line 191: Write “N. parvum” in italics, please.

Response: See response to Q2.

Q7. In line 204: Write “D. seriata” in italics.

Response: See response to Q2.

Q8. In line 217: Please write “ Botryosphaeria” in italics.

Response: Corrected.

Reviewer 4 Report

Dear Corresponding Author

Hope you are doing well. I checkedyour paper and I have some comments to improve the quality of your paper for publication:

1) You need a phylogenetic tree for identification of Botryosphaeriaceae fungi that you used in your experiment. DNA works are strongly required.

2) About non-commerial biocontrol isolates you need to have DNA works, too. 

3) In tables 2 and 3 in the part (mean of ranks) you need to have standard error. Instead of 'groups' part you have to have a bar-chart to present your results.

Regards

Round 2

Reviewer 1 Report

The authors did a good job of enhancing the brief report. I recommended accepting it after MPDI English editing as Brief Report .

Author Response

The authors did a good job of enhancing the brief report. I recommended accepting it after MPDI English editing as Brief Report.

Response: Given that the Reviewer has not pointed to any specific problems with the use of English, and that the other two Reviewers have not raised issues related to the English language and style, we have checked again the manuscript, trying to polish minor mistakes that had gone unnoticed in previous iterations.

Reviewer 2 Report

I appreciate the improvements to the manuscript and the answers by the Authors. The paper was notably improved and can be published by the Journal in the current form.

Author Response

I appreciate the improvements to the manuscript and the answers by the Authors. The paper was notably improved and can be published by the Journal in the current form.

Response: Thank you for your positive feedback.

Reviewer 4 Report

Dear Corresponding Author

You should add just one phylogenetic tree into the main manuscript not as suppl. mat. Bootstraps are low and your strain is probably a new species. Please clarify it to us. You need to use MrBayes method to inference a tree.

Standard error is not related to parametric or non-parametric data. You should add into a table or bar-chart.
